

**Mechanical State of Gravel Soil in Mobilization of Rainfall-Induced**
**Landslide in Wenchuan seismic area, Sichuan province, China**
**Liping Liao[1, 2, 3], Yunchuan Yang[1, 2, 3], Zhiquan Yang[4], Yingyan Zhu[5*] , Jin Hu[5], D.H.Steve Zou[6]**
[1]College of Civil Engineering and Architecture, Guangxi University, Nanning 530004, China
[2]Key Laboratory of Disaster Prevention and Structural Safety of Ministry of Education, Guangxi University, Nanning
530004, China
[3]Guangxi Key Laboratory of Disaster Prevention and Engineering Safety, Guangxi University, Nanning 530004, China
[4]Faculty of Land Resource Engineering, Kunming University of Science and Technology, Kunming 650500, China
[5]Institute of Mountain Hazards and Environment, Chinese Academy of Sciences and Ministry of Water Conservancy,
Chengdu 610041, China
[6]Department of Civil and Resource Engineering, Dalhousie University, Halifax, NS, Canada B3H4K5
Correspondence to: Y. Y. Zhu (zh_y_y_imde@163.com)
**Abstract** Although gravel soils generated by seismic shaking in Wenchuan earthquake area have subjected to natural
consolidation process for nearly ten years, geological hazards, such as slope failures with ensuing landslides, frequently
are haunting the area. In this paper, artificial flume model tests and triaxial tests were used to make close observation on
the mechanical state of gravel soil in Wenchuan seismic area. The results showed that: (1) The timing and patterns of
landslide initiations were closely related to their initial dry densities, and the initiation processes were accompanied with
a variation of dry density and void ratio; (2) Fine particle migration in soil and coarse-fine particle content rearrangement
contributed to the internal micro structure reorganization, which was supposed to be the main reason for variation of dry
density and void ratio; (3) Gravel soils with unchanged grain compositions, if under the same hydrostatic compression,
they approached to an identical critical void ratio to fail; (4) The mechanical state of certain sort of gravel soil can be
identified by its relative position between state parameter ($e$, $p'$) and $e_c$-$p'$ planar critical state line; (5) Gravel soil slope
failed and then evolved into landslide under lasting rainfall leaching, while in gravel slope there co-existed soil dilatation
and contraction, but the dilatation was dominant. Above research findings not only could be used to interpret landslide
initiation but also would provide an insight for landslide warning forecast of gravel slope in seismic area.
**Keywords** Mechanical state • gravel soil• landslide• critical state• Wenchuan seismic area



## 1    Introduction


In 2008, gravel soils generated by seismic shaking in Wenchuan earthquake area contributed to the large number of
loose deposits (Tang and Liang 2008; Xie et al., 2009). These deposits characterized by wide grading,
under-consolidation and low density, were locating at the both sides of highway and gully, and resulted in the
formation of soil slopes (Cui et al., 2010; Qu et al., 2012; Zhu et al., 2011). Although gravel soils have subjected to
natural consolidation process for nearly ten years, geological hazards, such as slope failures with ensuing landslides,
are readily to motivate when it suffers heavy rainfall and frequently are haunting the local region which caused
intense gully erosion, severe damages of the Duwen highway, and huge losses of life and property (Chen et al.,
2012a; Chen et al., 2017; Cui et al., 2013; Hu et al., 2016; Hu et al., 2014; Huang et al., 2012; Huang and Tang
2014; Li et al., 2010; Liu et al., 2016; Ma et al., 2013; Ni et al., 2014; Sun et al., 2011; Tang et al., 2012; Tang et al.,
2011b; Tang et al., 2009; Wang et al., 2015; Xu et al., 2012; Yin et al., 2016; You et al., 2012; Zhang and Zhang
2017; Zhang et al., 2013; Zhang et al., 2014; Zhou et al., 2015; Zhou and Tang 2013; Zhou et al., 2014; Zhuang et
al., 2012).

Fully understanding the mechanical state of gravel soil is an engineering and scientific basis for disaster
prevention and mitigation in a seismic area (Chen et al., 2010). Generally, the void ratio of soil is an important
parameter in describing the mechanical state quantitatively (Been and Jefferies 1985), which has already involved
the deterministic analysis of the critical state of soil, and belongs to an important branch of soil mechanics - the
critical state soil mechanics (Schofield and Wroth 1968).

The critical state soil mechanics indicated that soil must experience the transformation process of a relatively
steady condition into the critical state (Schofield and Wroth 1968). In 1936, Casagrande (1936) pointed out that the
critical state was that loose soil contracted, and dense soil dilated to the same critical void ratio in the drained
shearing test. Some of the observed phenomena of landslides might be approximately explained by the critical state
soil mechanics (Fleming et al., 1989; Sassa 1984; Wang and Sassa 2003), thus since the 1980s, the critical state of
soil had been introduced into the initiation mechanism of the landslide and debris flow, which had received
extensive attentions (Fleming et al., 1989; Sassa 1984; Verdugo and Ishihara 1996; Wang and Sassa 2003; Gabet
and Mudd 2006; Iverson et al., 2010; Iverson 2000; Iverson 2005; Iverson et al., 2000; Iverson et al., 1997; Schulz
et al., 2009). Wherein, in 1984, Sassa (1984) concluded that the liquefaction of loose sand was attributed to the
critical state; in addition, due to the incompressibility of water, the dilation and contraction behavior of the soil
resulted in the fluctuation of pore pressure in the undrained conditions. Based on the F line drawn by
Casagrande( 1936), in 1989, Fleming (1989) found that the increase of pore water pressure corresponded to the soil
dilation and the intermittent debris flow; however, his research was in contrast to the theory proposed by
Casagrande (1936) that "dilative soil was not easy to liquefy". In 1997, Iverson (1997) also found that the density
of loose sand increased, and the density of dense sand decreased to the same critical density. His research can
indirectly reflect the existence of the critical void ratio of soil. In 2000, Iverson et al (2000) demonstrated that in the
shearing process of soil, the contractive behavior of the loose loamy sand prompted pore pressure to increase
rapidly, which led to the immediate failure within the soil; in contrast, the dense loamy sand exhibited dilative
behavior which resulted in the decrease of pore pressures. He also pointed out although it was not easy to observe
the phenomenon that the landslide velocity depended on the void ratio, the void ratio played the important role in
the formation of landslide. In his paper, the value of the critical void ratio was not mentioned. As the critical void
ratio was a function of the mean effective stress (Verdugo and Ishihara 1996), based on the model of Iverson (2000)
and critical sate soil mechanics (Schofield and Wroth 1968), the theoretical formula of the critical void ratio is
deduced by Gabet and Mudd(2006). Besides, they pointed out the particle diameter of soils affected the rate at
which the soil reaches a critical state; when the rainfall duration is sufficient for the dense soil to reach the critical
void ratio, and to generate the excess pore pressure, the soil would dilate. This might be the reason for the



paradoxical conclusion made by Fleming (1989). William (2009) found out the dilative strengthening might control
the landslide velocity. In additon, other scholars also found that in the shearing process of soil, the critical state, the
dilative and contractive behavior was exiting in residual soil, loess and coarse grained soil (Dai et al., 2000; Dai et
al., 1999a; Dai et al., 1999b; Liu et al., 2012; Zhang et al., 2010; Zhu et al., 2005). Although the critical state soil
mechanics had been applied to explain the mobilization of landslide theoretically since 1980s (Dai et al., 2000; Dai
et al., 1999a; Dai et al., 1999b; Fleming et al., 1989; Gabet and Mudd 2006; Iverson et al., 2010; Iverson 2000;
Iverson 2005; Iverson et al., 2000; Iverson et al., 1997; Liu et al., 2012; Sassa 1984; Schulz et al., 2009; Verdugo
and Ishihara 1996; Wang and Sassa 2003; Zhang et al., 2010; Zhu et al., 2005), most precedent studies focus on the
qualitative results and lack the field testing data. In addition, the critical state of gravel soil in a seismic area is not
exactly identified in the field research. For example, is the mechanical state of gravel soil contraction or dilation?
How to estimate the mechanical state of gravel soil when the landslide initiates?
Through artificial flume model tests and triaxial tests, this paper investigates the mechanical state of gravel
soil in Niujuan valley, Yingxiu Town of Wenchuan County, Sichuan Province, China. More specially, first, the
variation of soil moisture content and pore water pressure, and the macro-micro property was observed. Second, the
mathematical expression of critical state of soil was proposed. Third, the mechanical state of gravel soil was
discussed.

## 89  2    Field site and method

### 90  2.1    Field site

Niujuan Valley is locating in Yingxiu town of Wenchuan County, Sichuan Province, which is the epicenter of 12 May
2008 Wenchuan earthquake in China. The main valley of the basin has an area of 10.46km$^2$, and a length of 5.8km. The
highest elevation is 2693m, and the largest relative elevation is 1833m. The range of the valley slope is 32.7%~52.5%
(Tang and Liang 2008; Xie et al., 2009). Six small ditches are distributing in the basin. The valley is characterized by the
abundant loose gravel soil, extremes of precipitous valley relief and the adequate rainfall, which contribute to the
frequent landslides and debris flows with large scale. Hence, this valley is regarded as the most typical basin in the
seismic area; and its excellent formative environment of landslide can provide the comprehensive reference model and
the rich soil sample for the artificial flume model tests.

### 99  2.2    Soil tests and quantitative analysis

#### 100  2.2.1    Artificial flume model test

Based on the field surveys along Duwen highway, Niujuan valley and the literature review (Chen et al., 2010; Fang et al.,
2012; Tang et al., 2011a; YU et al., 2010), most of rainfall induced landslides is the shallow landslides, and the range of
the slope gradient is 25°~40°; besides, the cumulative content of silt and clay (particle diameter < 0.075mm) is about 2%,
which plays the important role in the mobilization of landslide and debris flow (Chen et al., 2010); the rainfall intensity
triggering the landslides is 10mm/h~70mm/h. Considering the above basic data, the authors designed the artificial flume
model, as shown in Fig. 1 (a). The length, width and height of the flume model are 300cm, 100cm and 100cm
respectively.
The gravel soil samples are from Niujuan valley (Fig. 1 (b)). The specific gravity is 2.69. The minimum, the
maximum dry density is 1.48g/cm$^3$ and 2.36g/cm$^3$; in addition, the minimum, the maximum void ratio is 0.14 and 0.82.
The grading curve is shown in Fig. 1 (c). As shown in Fig. 1(c), the cumulative content of gravel (particle diameter <
2mm) is 30.74%, and the cumulative content of silt and clay (particle diameter < 0.075mm) is 2.78%. The model tests
comprise of four initial dry densities: 1.54g/cm$^3$, 1.62g/cm$^3$, 1.72g/cm$^3$, 1.81g/cm$^3$ (Tab. 1), because the initial dry
density influenced the formation of landslide (McKenna et al., 2011). In order to achieve a predetermined initial dry
density, the soils of the models are divided into four layers and compacted. The thickness of each layer is 20cm, 15cm,



15cm and 10cm respectively (Fig. 1 (a)).

Artificial rainfall system, which was designed by the Institute of Soil and Water Conservation, CAS, comprises of

two spray nozzles, a submersible pump, water box and a bracket. The range of nozzle sizes is 5~12mm, thus, the actual
rainfall intensity in the field can be simulated. Three groups of sensors, including the micro-pore pressure sensors (model:
TS-HM91, produced in England) and moisture sensors (model: SM300, produced in England), are placed between two
layers of the soil to measure the volume water content and the pore water pressure (Fig. 1 (a)). DL2e device is applied to
collect the data from the sensors, which can scan 30 channels within the same second, so that the time interval of data
collection is set to one second.
2.2.2    Triaxial test
Triaxial tests were carried out on the dynamic triaxial apparatus in Institute of Mountain Hazards and Environment, CAS.
The diameter and the height of sample were 15 cm and 30 cm (Fig. 2). The test is the saturated and consolidated drainage
shear test at a shear rate of 0.8mm/minute, which comprise of two sets: the initial dry density of 1.94 and 2.00g/cm$^3$. The
confining pressure is 50Kpa, 100Kpa and 150Kpa.

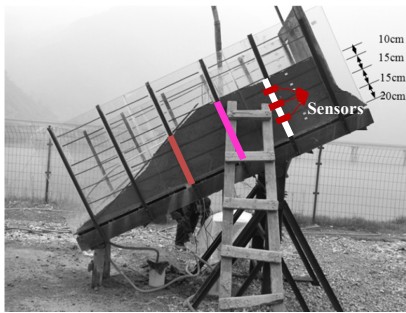 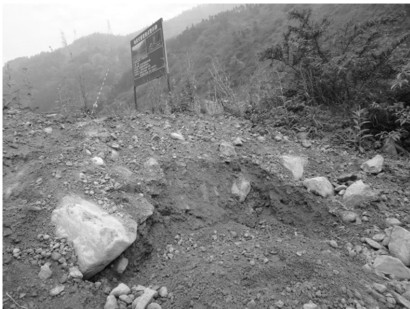


(a)    Artificial flume model (the position of sampling: red line-1#, pink line-2#, white line-3#) (b) Gravel soil in Niujuan valley

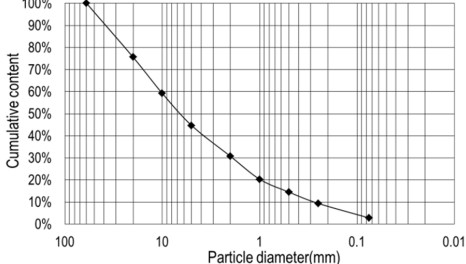


(c) Grain composition of gravel soil particle

**Fig. 1** Test model and grain composition of gravel soil particle
**Tab. 1** Sets of artificial flume soil model test

| Factor / Number | Initial mass moisture content (%) | Gradient of slope (°) | Rainfall intensity (mm/h) | Initial dry density (g/cm$^3$) |
|---|---|---|---|---|
| 1 |  |  |  | 1.54 |
| 2 | 6~8 | 27 | 47~50.2 | 1.62 |
| 3 |  |  |  | 1.72 |
| 4 |  |  |  | 1.81 |




**Fig. 2** Triaxial test equipment
2.2.3    Quantitative analysis method
Quantitative analysis is mainly based on artificial flume model test and triaxial test. Firstly, the state parameters of soil
are expressed by the void ratio $e$ and the mean effective stress $p'$, which can be derived from the artificial flume model
test. In artificial flume model test, at least three soil samples are collected by soil sampler in the same depth of the line 1#,
2# and 3#, and are used to calculate their natural density $\rho$, mass moisture content $\omega$ and dry density $\rho_d$. Later, void ratio
can be calculated by the formula: $e=G_s/\rho_d -1$ ($G_s$ is the specific gravity). The cumulative content of coarse $P_5$ (particle
diameter > 5mm), gravel (particle diameter < 2mm) $P_2$, and silt and clay (particle diameter < 0.075mm) $P_{0.075}$ is obtained
from the particle grading test. The mean effective stress $p'$ is equal to one third of the sum of $\sigma_x$, $\sigma_y$ and $\sigma_z$, wherein, the
vertical stress $\sigma_z$ is equal to $\gamma h$, the horizontal stress $\sigma_x$ and $\sigma_y$ is equal to $K_a\gamma h$; $h$ is the vertical distance between the some
point inside the slope and the surface of the slope; $\beta$ is the gradient of the slope; $\gamma$ is the soil bulk density; $K_a$ is the lateral
pressure coefficient, which can be calculated by the formula (1) (Chen et al., 2012b); $\phi$ is the internal friction angle of
soil. In this paper, $\beta$=27°, $\phi$=33°.
$$K_a = \cos\beta \frac{\cos\beta - \sqrt{\cos^2\beta - \cos^2\phi}}{\cos\beta + \sqrt{\cos^2\beta - \cos^2\phi}} \tag{1}$$

Secondly, the critical state line (CSL, $e_c$-ln$p'$) is derived from the saturated and consolidated drainage shear test.
Finally, based on the critical state soil mechanics, according to the relative position of the state parameter ($e$, $p'$) at the
CSL, the mechanical state of the soil can be estimated. When the soil state ($e$, $p'$) is located at the upper right of the CSL,
the soil is contracted. When the soil state ($e$, $p'$) is located at the lower left of the CSL, the soil is dilated (Casagrande A
1936; Schofield and Wroth 1968).
**3    Results**
**3.1    Soil moisture content and pore water pressure**
As shown in Fig.3~Fig.6, the variation of the volume moisture content of soil depth of 10~25cm exhibits the similar
tendency, which includes the constant state since the beginning of rainfall, the rapid increase when the rainfall seeping
into soil, and the steady growth trend at the end. However, throughout the rainfall, the volume moisture content of soil
depth of 40cm exhibits a slow-growth trend or remains the stable. For example, when the dry density is 1.54g/cm³, at the
beginning of rainfall, the volume moisture contents of three depths all remain the same. When the rainfall duration is
about 500s, the volume moisture content of soil depth of 10cm begins to increase fast, while the volume moisture content
of soil depth of 25cm~40cm still remains unchanged. When the rainfall duration is about 1200s, as the rainfall penetrates
into the internal soil, the volume moisture content of soil depth of 25cm begins to increase suddenly, while the volume
moisture content of soil depth of 10cm maintains a slow changing trend. Besides, the variation of pore water pressure
shows the similar trend which is characterized by a sharp increase at first, then decreases rapidly and the continuous
dynamic fluctuation. The pore water pressure of soil depth of 10~25cm is mostly positive, while when the initial dry
density is 1.81g/cm³, the pore water pressure of soil depth of 40cm changes from positive to negative when the rainfall



duration is about 6000s.

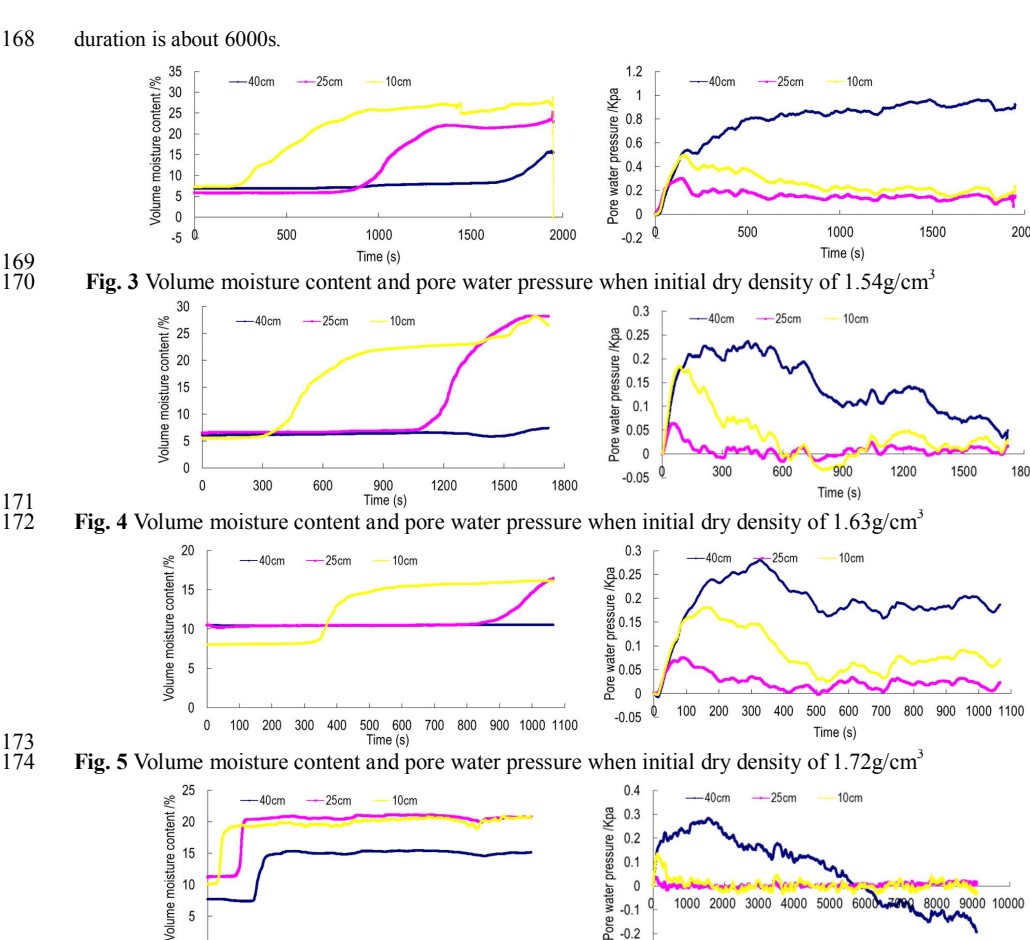

**Fig. 3** Volume moisture content and pore water pressure when initial dry density of 1.54g/cm$^3$
**Fig. 4** Volume moisture content and pore water pressure when initial dry density of 1.63g/cm$^3$
**Fig. 5** Volume moisture content and pore water pressure when initial dry density of 1.72g/cm$^3$
**Fig. 6** Volume moisture content and pore water pressure when initial dry density of 1.81g/cm$^3$
**3.2    Macro and micro property of gravel soil**
When the initial dry density of gravel soil is 1.54~1.72g/cm$^3$, the landslide can be triggered by rainfall; however, the
initiating processes of landslides have their similarity and difference. The similarity is that at the beginning of rainfall, the
shallow soil is compacted due to the seepage force of rainfall (Fig. 7 (a)). In addition, surface runoff cannot be observed
during the rainfall duration, while the muddy water can be generated and overflow the slope foot (Fig. 7 (b)). This
phenomenon indicates that all the rainfall can seep into the internal soil; the fine particles (mainly clay and silt) along the
seepage paths start to migrate and are attributed to the formation of the subsurface flow inside the slope. This migration
process results in the variation and redistribution of the soil micro-structure (Chen et al., 2004; Zhuang et al., 2015). The
difference of initiating process is that the time and pattern. For example, when the initial dry density is 1.54~1.63g/cm$^3$,
the initiating time of landslide is 30~40 minutes. The steps of landslide initiation are as follows: first, the soil of the
superficial layer slowly slides in the shape of soil flow (Fig.8 (a)); second, a small-scale slip occurs (Fig.8 (b)); third, the
large-scale slide of soil is motivated (Fig.8 (c)). When the initial dry density is 1.72g/cm$^3$, the initiating time of landslide
is 18 minutes. Before landslide initiation, firstly, the shear opening occurs accompanied by the cracks developing in the
slope foot (Fig. 9 (a)); secondly, some cracks develop inside the top of the slope (Fig. 9 (b)); finally, landslide initiates



accompanied by the instantaneous expansion of cracks (Fig. 9 (c)), which takes 5s. This initiation process implies that
when the fine particles migrate, the particles in the framework start to move in translation and rotation under the action of
gravity, and fill the interval space and block the downstream channels of the seepage path. All the above process can lead
to the decrease of the void ratio and the increase of the pore water pressure, and result in the formation of sliding fracture
surface (Gao et al., 2011). When the initial dry density is 1.81g/cm$^3$, the slope keeps stable and landslide cannot be
triggered by the rainfall even though the fine particles disappear, and the coarse particles are exposed at the slope surface.

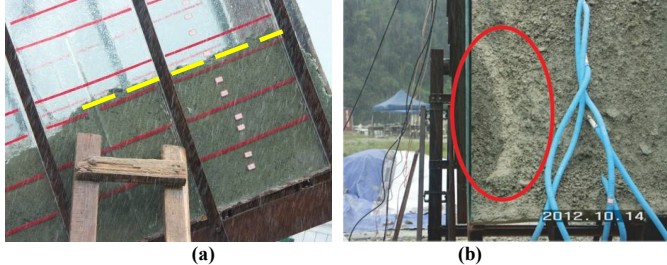

**(a)**             **(b)**
**Fig.7** Similarity of process of landslide initiation

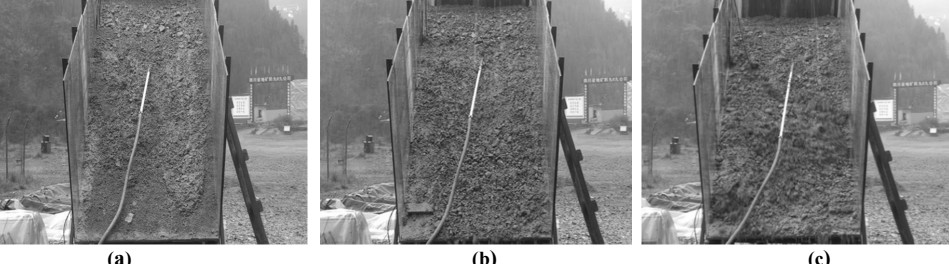

**(a)**         **(b)**         **(c)**
**Fig. 8** Process of landslide initiation (initial dry density of 1.54~1.63g/cm$^3$)

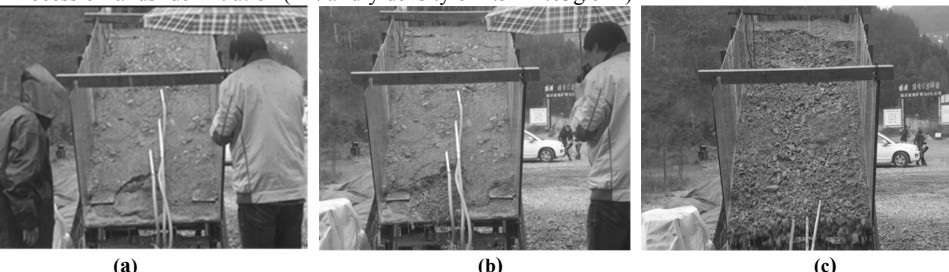

**(a)**         **(b)**         **(c)**
**Fig. 9** Process of landslide initiation (initial dry density of 1.72g/cm$^3$)
As shown in Tab.2, when the initial dry density is 1.54~1.72g/cm$^3$, the natural density and dry density of soil depth
of 5cm~20cm (line 1#, 2# and 3#) are larger than those before the test, and the void ratio is smaller than it before the test.
Among these three lines, the rate of change of natural density and dry density in line 1# is the highest. When the initial
dry density is 1.63cm$^3$, the dry density of soil depth of 40cm (line 3#) is smaller than it before the test, and the void ratio
is larger than it before the test. When the initial dry density is 1.81g/cm$^3$, the natural density and dry density increase after
the test.





**Tab 2.** Density and void ratio of gravel soil with initial dry density 1.54~1.81g/cm$^3$

| Number | Initial dry density (g/cm$^3$) | Line number | $h$ (cm) | Natural density of soil close to failure $\rho$ (g/cm$^3$) | Mass moisture content $\omega$ (%) | Dry density of soil close to failure $\rho_d$ (g/cm$^3$) | Void ratio close to failure $e=G_s/\rho_d-1$ | $\sigma_z=\gamma h$ (Kpa) | $\sigma_x=\sigma_y=K_a\gamma h$ (Kpa) | $p'=(\sigma_x+\sigma_y+\sigma_z)/3$ (Kpa) |
|---|---|---|---|---|---|---|---|---|---|---|
| 1 | 1.54 | 3# | 5 | 2.08±0.05 | 9.35±0.85 | 1.90±0.04 | 0.39±0.03 | 1.04 | 0.59 | 0.74 |
| | | 3# | 28 | 1.93±0.03 | 8.61±1.16 | 1.77±0.02 | 0.49±0.02 | 5.39 | 3.07 | 3.84 |
| | | 2# | 33 | 2.07±0.05 | 9.15±0.15 | 1.89±0.04 | 0.40±0.03 | 6.82 | 3.88 | 4.86 |
| | | 1# | 21 | 2.10±0.05 | 9.63±1.01 | 1.91±0.05 | 0.39±0.04 | 4.40 | 2.51 | 3.14 |
| 2 | 1.63 | 3# | 5 | 2.19±0.01 | 13.36±0.09 | 1.98±0.01 | 0.34±0.01 | 0.44 | 0.25 | 0.31 |
| | | 3# | 40 | 1.67±0.03 | 6.15±0.17 | 1.58±0.02 | 0.68±0.02 | 6.68 | 3.80 | 4.76 |
| | | 2# | 20 | 2.09±0.04 | 10.18±0.21 | 1.90±0.04 | 0.39±0.03 | 4.19 | 2.38 | 2.99 |
| | | 1# | 13 | 2.23±0.04 | 10.84±0.83 | 2.01±0.02 | 0.32±0.02 | 2.90 | 1.65 | 2.07 |
| 3 | 1.72 | 3# | 10 | 2.22±0.02 | 8.45±0.72 | 2.05±0.02 | 0.30±0.01 | 2.22 | 1.26 | 1.58 |
| | | 3# | 25 | 2.34±0.04 | 8.59±0.261 | 2.16±0.05 | 0.23±0.03 | 5.86 | 3.33 | 4.17 |
| 4 | 1.81 | 1# | 10 | 2.30±0.01 | 9.26±0.42 | 2.10±0.01 | 0.26±0.01 | 2.30 | 1.31 | 1.64 |
| | | 3# | 5 | 2.14±0.04 | 9.57±0.75 | 1.95±0.04 | 0.36±0.03 | 1.28 | 0.73 | 0.91 |
| | | 3# | 10 | 2.26±0.01 | 8.16±0.39 | 2.09±0.02 | 0.27±0.01 | 2.26 | 1.28 | 1.61 |

As shown on the section 2.2.1, $P_5$ of soil before the test is 55.32%; therefore, the coarse particles and the fine
particles interact with each other to form the soil structure, which influence the changes of the dry density (Guo 1998)
and landslide or debris flow characteristics (Li et al., 2014). In order to find out the reasons for the variation of dry
density and void ratio, $P_5$, $P_{0.075}$ and $P_2$ of line 1# and 3# before and after tests are compared. As shown in Tab.3, in the
condition of the same initial dry density, when the initial dry density is 1.54g/cm$^3$ and 1.63g/cm$^3$, the loss of $P_{0.075}$ is the
largest in the shallow layer of the slope top, followed by the loss of $P_{0.075}$ in the slope foot. It is indicated that in the early
period of rainfall, at the slope top, the fine particles of the shallow soil mainly migrate along the direction of gravity;
when the interflow forms, the fine particles begin to move to the slope foot. This process results in the porosity at the
migration position increases, while the porosity of the position which is filled by fine particles decreases (Wang et al.,
2010). It is also regarded as the seepage-compacting effect (Jiang et al., 2013). As a result, the shallow soil on the slope
top is looser than the shallow soil on the slope foot. The loss of $P_{0.075}$ at the slope top decreases significantly with depth.
Especially, it is about -1.26% at the depth of 40cm. It is indicated that the depth of rainfall infiltration is about 40 cm.
When the range of the initial dry density are 1.72g/cm$^3$~1.81g/cm$^3$, with the increase of depth, the variation of $P_{0.075}$ at
the slope top changes from negative to positive. This trend indicates that the fine particles migrate and deposit at the
depth of 5~25cm wherein the depth is 10~25cm, 5~10cm for the initial dry density of 1.72g/cm$^3$ and 1.81g/cm$^3$.
**Tab 3.** Variation of coarse and fine particles contents

| Number | Initial dry density (g/cm$^3$) | Line number | $h$ (cm) | $P_5$ | $\Delta P_5$ | $P_{0.075}$ | $\Delta P_{0.075}$ | $P_2$ | $\Delta P_2$ |
|---|---|---|---|---|---|---|---|---|---|
| 1 | 1.54 | 3# | 5 | 61.00% | 10.25% | 0.66% | -76.24% | 30.69% | -0.16% |
| | | 3# | 28 | 55.91% | 1.05% | 2.01% | -27.90% | 34.36% | 11.76% |
| | | 1# | 21 | 58.98% | 6.60% | 0.77% | -72.36% | 31.07% | 1.05% |
| 2 | 1.63 | 3# | 5 | 58.69% | 6.09% | 0.91% | -67.23% | 31.40% | 2.15% |
| | | 3# | 40 | 57.98% | 4.80% | 2.75% | -1.26% | 31.69% | 3.07% |
| | | 1# | 13 | 67.66% | 22.30% | 1.26% | -54.81% | 26.23% | -14.68% |
| 3 | 1.72 | 3# | 5 | 55.98% | 1.18% | 1.03% | -62.98% | 32.70% | 6.38% |
| | | 3# | 10 | 54.01% | 2.37% | 1.78% | -36.14% | 33.94% | 10.40% |
| | | 3# | 25 | 55.32% | 0% | 3.17% | 13.85% | 34.05% | 10.75% |





| | | 1# | 10 | 56.15% | 1.5% | 1.42% | -49.09% | 33.67% | 9.53% |
|---|---|---|---|---|---|---|---|---|---|
| 4 | 1.81 | 3# | 5 | 52.50% | -5.11% | 2.06% | -25.83% | 35.49% | 15.45% |
| | | 3# | 10 | 52.55% | -5.01% | 2.86% | 2.68% | 33.91% | 10.30% |

Note: the positive value of the change represents an increase while the negative value represents a decrease.
On the slope top, the trend of $P_5$ (the depth of 5cm) is from positive to negative with the increasing of initial dry
density, which range is from -5.11% to 10.25%. The reason is that the loss of fine particles contributes to the relatively
increase of the coarse particles' content. The overall trend of $P_{0.075}$ at the slope top and slope foot both decreases, which
range is from 25.83% to 76.24% and from 49.09% to 72.36% respectively. The relationship between the loss of $P_{0.075}$
($\Delta P_{0.075}$, which is negative) at the slope top and slope foot and initial dry density $\rho_d$ is shown in Fig. 10. The regression
equation is as follows: $\Delta P_{0.075}=1.2632\rho_d - 2.6464$, $\Delta P_{0.075}=1.709\rho_d - 3.4391$, and $R^2$ is 0.8827, 0.8199 respectively. It is
indicating that $\Delta P_{0.075}$ has the significant correlation with $\rho_d$; specially, the greater initial dry density, the smaller loss of
$P_{0.075}$. When $\rho_d$ is 1.53g/cm³, $P_2$ decreases and the amount of change is -0.16%, while $\rho_d$ is 1.63~1.81g/cm³, $P_2$ increases
with the range of 2.15%~15.45%. The reason for the loss of $P_{0.075}$ and $P_2$ is that the fine particles, including silt and clay,
are continuously motivated to move by the subsurface flow. The reason for the increase of $P_2$ might be that during the
rainfall, the large gravels keep rolling off, so that the content of particles larger than 2 mm decreases, so that the content
of particles smaller than 2 mm relatively increases.

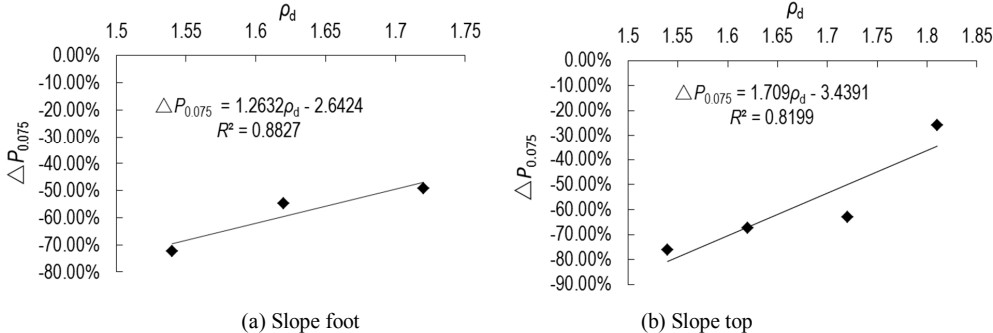


(a) Slope foot                                    (b) Slope top

**Fig. 10** Relationship between $\Delta P_{0.075}$ and $\rho_d$
### 3.3    Critical state of gravel soil
**(1)    Definition of critical state and calculation of porosity ratio**
Casagrande et al pointed out under the condition of continuous shear load, the constant state of the deviation stress
and the void ratio are the critical state (Casagrande A 1936; Roscoe et al., 1963; Schofield and Wroth 1968). For the
consolidation drainage test, under a certain confining pressure, as the axial strain $\varepsilon_a$ increases, the principal stress $q$ and
the volume strain $\varepsilon_v$ tends to a stable value, at this time the soil is in a critical state characterized by the plastic flow (Liu
et al., 2011). According to the results of triaxial shear tests, when the axial strain reaches 16%, the deviation stress is
stable, and the absolute value of the increment of volume change to the current volume change is less than 0.01; the soil
enters the critical state (Liu et al., 2012). A certain relationship between the void ratio and the volumetric strain exists in
sand and gravel soil, so that the current porosity ratio $e$ is calculated by formula (2) (Xu et al., 2009), wherein $e_0$ is the
initial void ratio.

$$e = \left(1+e_0\right)\exp\left(-\varepsilon_v\right)-1$$                            (2)

**(2)    The critical line in the $e_c$- $p'$ plane**
Tab. 4 shows the critical void ratio $e_c$, $q$ and $p'$ under two initial dry densities. As shown in Table 4, the same critical
void ratio will be reached approximately for the gravel soil with initial dry density of 1.94g/cm³ and 2.00g/cm³. This
result is consistent with existing study (Gabet and Mudd 2006; Iverson et al., 2000), which can indicate that gravel soil




also has the similar principle that the soil with the same grade will shear to reach the same critical void ratio.
**Tab 4.** Critical void ratio $e_c$ of gravel soils

| $\sigma_3$(Kpa) | Initial dry density (g/cm³) | $e_c$ | $q$ (Kpa) | $p'$ (Kpa) |
|---|---|---|---|---|
| 50 | 1.94 | 0.32 | 93.41 | 95.98 |
| | 2.00 | 0.34 | 69.50 | 84.65 |
| 100 | 1.94 | 0.30 | 227.43 | 213.80 |
| | 2.00 | 0.30 | 159.14 | 178.13 |
| 150 | 1.94 | 0.27 | 324.79 | 312.39 |
| | 2.00 | 0.29 | 181.12 | 239.86 |

The fitting curve of $e_c$ and $\ln p'$ is shown in Fig. 11 (a). The correlation coefficient is 0.8566, which indicates a statistically
significant relationship between $e_c$ and $p'$. According to the normalized residual probability, P-value of 0.964 is greater than the
selected significance level (P=0.05), which indicates that the residuals follow a normal distribution. Therefore, the mathematical
expression of $e_c$-$\ln p'$ of gravel soil in the critical state is as follows:
$$e_c = 0.5241 - 0.04304 \ln p' \tag{3}$$
**(3)   The critical line in the $q$- $p'$ plane**
The fitting curve of $q$ and the $p'$ is shown in Fig. 11 (b). The correlation coefficient is 0.9465, which indicates a statistically
significant relationship between $q$ and $p'$. The mathematical expression of $q$- $p'$ is as follows:
$$q = 0.6641(p')^{1.063} \tag{4}$$

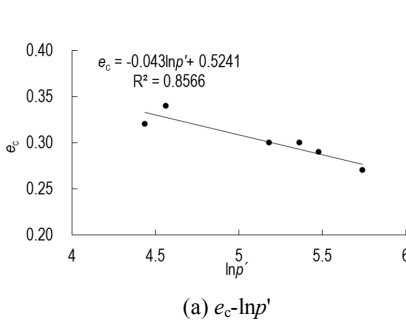
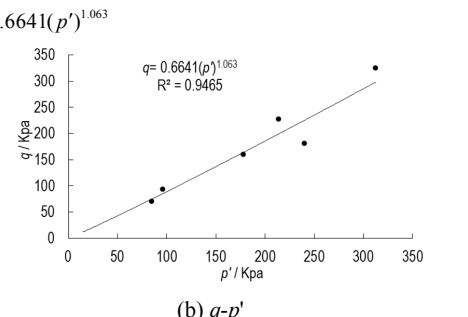


(a) $e_c$-$\ln p'$          (b) $q$-$p'$
**Fig. 11** Critical sate line of gravel soil
**4   Discussion**
The relative position of the state parameter ($e$, $p'$) at the critical state line is shown in Fig.12. As shown in Fig.12, when
the initial dry density is 1.54g/cm³~1.63g/cm³, the shallow soil at the slope top, the soil inside the middle of the slope and
at the slope foot dilates, while the soil with the depth 28cm and 40cm at the slope top contracts. When the initial dry
density is 1.72g/cm³~1.81g/cm³, the soil on the slope top and slope foot both dilate. When the initial dry density is
1.81g/cm³, the landslide cannot be triggered by the rainfall. The reasons might be that the soil is in the dense state;
therefore, the permeability capacity of the soil is low and the infiltration depth is restricted; the loss of the pore water
pressure due to soil dilation is difficult to recover timely, which can result in the discontinuity of the shear deformation.
The results show that there are two types of dilation and contraction in the mechanical state of gravel soil when the
landslide initiates; specially, dilation is the primary type.





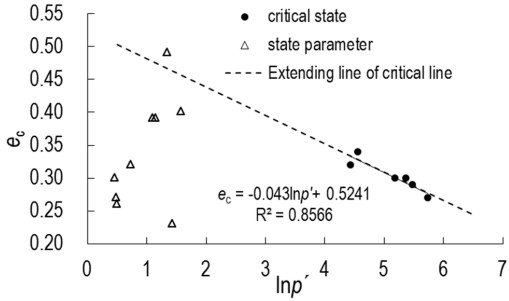


**Fig.12** Mechanical property of gravel soil

During the initiating process of landslide, the gravel soil slope changes from the unsaturated state to the saturated state. Due to the increase of pore water pressure, the soil of potential sliding surface falls into the shear failure under the drainage condition; however, the shear strain is small. When the soil is damaged by shear shrinkage, the porosity decreases after the soil is destroyed; the excess pore water pressure cannot be quickly dissipated in a short period of time, which causes the pore water pressure of the soil near the damage position to increase, and contributes to the decrease of the mean effective stress. The whole process of landslide initiation exhibits a sudden characteristic.

When the soil is destroyed by shear dilation, rainfall infiltration leads to the increase of pore water pressure in the soil near the potential slip surface; a small part of soil at the slope foot begins to slip, which causes the sliding force increase; subsequently, the effective stress decreases and the shear deformation occurs. At this moment, the pore water pressure decreases; the loss of shear strength due to strain softening is restored, and the deformation of soil is stopped. If there is the sufficient infiltration of rainfall, the pore water pressure can be recovered, and the soil deformation can continue. When the soil is in a dense state (relative density $D_r > 2/3$), if the infiltration rate is less than the rainfall intensity, it is difficult for the soil to reach the critical state due to short-term rainfall; at this moment, the slope still keeps stable. The macroscopic phenomenon of the soil deformation is a kind of local deformation and destruction, such as circumferential cracks, partial collapse or uplift. If the infiltration rate is larger than the rainfall intensity, although the rainfall infiltration is enough to break the mechanical balance of slope, its process still needs a relatively long period of time, so the macroscopic deformation soil appears as a gradual deformation and damage, such as the multiple slides and landslide. When the soil is in a medium dense state ($1/3 < D_r \leq 2/3$), the loss of the pore water pressure due to dilation will be recovered because of the rapid infiltration of rainfall, the shear deformation of soil will continue. The macroscopic phenomenon of soil deformation will appear as a kind of sudden failure (Dai et al., 2000).

## 5   Conclusion

(1)   The timing and patterns of landslide initiations were closely related to their initial dry densities, and initiation processes were accompanied by a variation of dry density and void ratio. The overall trend is that the dry density at the depth of 5cm~20cm increases, and the void ratio decreases. The change rate at the slope foot is the largest. When the initial dry density is 1.63g/cm$^3$, the dry density (the depth of 40cm on the slope top) decreases, and its porosity increases.

(2)   Fine particle migration in soil and coarse-fine particle content rearrangement contributed to the internal micro structure reorganization, which was supposed to be the main reason for variation of dry density and void ratio. When the initial dry density is 1.54g/cm$^3$ and 1.63g/cm$^3$, the variation of $P_{0.075}$ (the depth of 5cm at the top of the slope) is the largest, followed by the variation of $P_{0.075}$ at the slope foot. The variation trend of $P_5$ changes from increasing to decreasing. The loss of $P_{0.075}$ at the top of the slope decreased significantly with depth; in addition, the loss of $P_{0.075}$ at the slope top the and at the slope foot both have the positive correlation with the initial dry density. $P_2$ has the increase trend for the initial dry densities of 1.63~1.81g/cm$^3$ except 1.54g/cm$^3$.

(3)   The same critical porosity ratio will be reached approximately for the gravel soil with initial dry density of 1.94g/cm$^3$ and 2.00g/cm$^3$.



(4) The mathematical expression of $e_c$-ln$p'$, $q$-$p'$ of gravel soil in the critical state is as follows:
$e_c$=0.5241-0.04304ln$p'$, $q$=0.6641$(p')^{1.063}$.
(5) The relative position of the state parameter $(e, p')$ at the critical *state* line is applied to estimate the mechanical
state of gravel soil.
(6) There are two types of dilation and contraction in the mechanical properties of gravel soil when landslide
initiates. In addition, dilation is the primary type.
**Acknowledgements**
This study was funded by the National Natural Science Foundation of China (No 41071058, 41402272, 51609041);
Disaster Prevention and Mitigation and Engineering Safety Key Laboratory Project of Guangxi Province (No
2016ZDX09).

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
