# Peer review of "Mechanical State of Gravel Soil in Mobilization of Rainfall-Induced"

_Earth Surface Dynamics, 2018_

## Referee Comment (RC1) · Anonymous Referee #1 · 18 Apr 2018

General comments This paper presented a study about the mechanical state of gravel soil in the landslide initiation using artificial flume model tests and triaxial tests. This topic is very interesting and significant for the landslide early identification and prediction, and it is within the scope of ESURF. The experiment and testing are designed reasonably and its results are reliable. but I think the innovation of this paper is slightly weak. The Introduction and Conclusion did not prepare well. In addition, the language of this paper should be improved. I think this paper needs a round of major revision before publication. Specific comments 1. I think the introduction was not prepared well. too many previous studies were presented, only important studies related to you study should be presented; the purpose and motivation of this paper should be clearer. 2.

[Figure]

The initial dry density is important for the analysis and conclusions, I suggest the authors add some explanation that why or how these four initial dry densities (1.54g/cm3, 1.62g/cm3, 1.72g/cm3, 1.81g/cm3) were selected? 3. In the Section of 3.1, the authors stated that 'throughout the rainfall, the volume moisture content of soil depth of 40cm exhibits a slow-growth trend or remains the stable'; however, as shown in Fig. 6, the volume moisture content of soil depth of 40cm increased sharply, please provide a brief explanation for this phenomenon. 4. The authors design the experiment to explore the relationship between the initial dry density and landslide initiation. With the results, it was proved that they have a very close relationship. But, it is still not clear that what the relationship is. For example, why the initiating time of the landslide with the initial dry density of 1.72g/cm3 (18 minutes) is shorter than the landslide with the initial dry density of 1.54-1.63g/cm3 (30 40 minutes). A deep analysis is needed. 5. In the Section of Critical state of gravel soil, the gravel soil with an initial dry density of 1.94g/cm3 and 2.00g/cm3 were used, why not the soil sample used before (1.54-1.81g/cm3)? 6. In my opinion, the conclusion section was not written well. the 5th conclusion is not clear; I suggest the conclusions about the Critical state of gravel soil can be synthesized. Technical corrections Line 36-41: please cite only the important references, it is unnecessary to list all the related literature; Line 91: I suggest the authors provide a location map with Niujuan Valley and Duwen highway. Line 93: please check the unit of '32.7Line 116: what does 'CAS' mean, please provide its definition. Line 120: what does 'DL2e' mean; Line 166-167: please correct the sentence; Line 240-241: please check the langue; Tab 2: please check the value of initial dry density, 1.62 or 1.63? it is not clear the meaning of h(cm), soil depth? Tab 3: it is not clear the meaning of âŰşP0.0075, âŰşP5, âŰşP2 and h. Tab 4: please provide the definition of $\sigma 3$; Fig.7-9: please add captions for each subfigure;

---

## Referee Comment (RC2) · Anonymous Referee #2 · 11 May 2018

With flume and triaxial tests, this paper investigates the mechanical state of gravel soil in Niujuan valley, Sichuan, China. The authors mentioned that they observed the variation is soil moisture content and pore water pressure, and the macro-micro property. They said to have presented a mathematical expression of critical state of soil. And finally discuss the mechanical state of gravel soil. The topic is very interesting.

However, no new mathematical formulation and model appeared in the text, except for some regression fits. There are several inconsistent statements. Lots of data are presented, great job! But, with much less insights and implications. Both the quality of science and presentation is poor. About 1/2 of the MS is very quantitative and

geotechnical, while another 1/2 is very descriptive. How do you relate these data to field events? What are the implications for the surface flow process and run-out modelling? These are not very strongly connected. Large part of the manuscript would perhaps better fit to some geotechnical and civil engineering journals than E-Surf. E.g. L137-153; L191-312. Probably these data would be interesting more to geotechnicians, and perhaps less to the audience of earth surface process. Otherwise, strongly justify how this is not the case. The journal and the Editors can decide on it.

The font size is too small. It was very difficult for me to read the print even with the power glasses. Time to time there are > 25 citations at a place! What is the use/purpose of this? This is fully distracting! Why don't you properly utilize the space for useful science/research? I thought the Journal/Editor should also have some initial controls on these and other aspects, at least the basic quality and content of the manuscript, before it is sent for reviews.

English, in general is good, but time to time difficult to follow, often strange, and needs to be substantially improved.

Detailed and critical comments:

L23: "state parameter ..." : The audience would not know this here without explaining what they are.

L26: "forecast": It is not clear, also in the main text, how you could forcast, what does it mean? Can you predict cracks formation and propagation, time, location and scale for forcasting and warning? No method is presented for this. If possible, please explain clearly how you could do that with the data and the models you are discussing.

L31,32: Improve English (ENG.). E.g., were locating –> were located, etc.

L36-41: There are > 25 citations here! What is the use/purpose of this? I would suggest to reduce it to about 3.

L42: "Fully understanding": Never possible. Improve writing.

L42-46: Looks like introductory undergraduate text.

L50: "Some of the observed phenomena of landslides": Not clear which?

L53-55: Again, so may citations. Do you need all these at once? Limit to about 3.

L57: Readers would under at this point what F is?

L59: "the intermittent debris flow": what is it?

L60-69: Strange writing. Unnecessary details, some irrelevant, not connected.

L74: "landslide velocity": Which velocity? Initiation, or dynamical until runout? You did not present data and analysis for velocity. Also, the dynamic velocity would, at most, negligibly depend on the initial state you are referring to. Otherwise, present data and analysis to support your arguments.

L75-80: Again, > 25 citations at one place. This is fully distracting! Why don't you properly utilize the space for useful science/research?

L81-80: "the critical state of gravel soil in a seismic area is not exactly identified in the field research": Why does it matter if it is sesimic or not?

L96, 102: "large scale", "most of rainfall induced landslides is the shallow landslides": inconsistent presentations. What is large scale?

L103-104: "silt and clay (particle diameter < 0.075mm) is about 2\%, which plays the important role in the mobilization of landslide and debris flow": How? Without proof and discussion, statements are useless.

L119: "produced in England": Do you need to say this? Why not to use reference properly?

L127-129: Fig. 1a: Initial shape and wedge angle needs to be discussed, also why chosen this way?

L143: "The mean effective stress p' is equal to one third of the sum of $\sigma x$, $\sigma y$ and $\sigma z$":

Do you really need to say this? There are lots of unnecessary things, making the MS much less professional.

L145: "is the soil bulk density": No!

L156-160: Eng.

L168-175: The yellow lines in Fig. panels cannot be seen. Better, plot in different line styles. Explain why the yellow lines are mostly in between the other lines on the right panels? All panels must be plot for the same x- and y-labels for better comparison. The mechanical and geotechnical reasons for the spacial behaviors seen in these panels are not well explained. Furthermore, how these behaviors influence dilation, landslide initiation, velocity and run-out?

L178: "the landslide can be triggered by rainfall": Show the hydro-mechanical relationship with the above figure. Otherwise, what is the use of the above data?

L184-185: Eng.

L185-186: "For example, when the initial dry density is 1.54∼1.63g/cm 3, the initiating time of landslide is 30∼40 minutes.": You must relate this with Fig. 6, right panels. No insight about the mechanics and process are mentioned, linked, and discussed. Otherwise, what is the use of Fig. 6?

L191: "expansion of cracks": Show it and the dynamics.

L192: "and rotation": how, where do you see it?

L193-194: "All the above process can lead to the decrease of the void ratio and the increase of the pore water pressure": Not clear how?

L195-196: "When the initial dry density is 1.81g/cm 3 , the slope keeps stable and landslide cannot be triggered by the rainfall even though the fine particles disappear, and the coarse particles are exposed at the slope surface.": This is important. Explain with strength relation.

L196-205: The figure captions don't explain the process in panels, difficult to follow.

L252-254: Difficult to follow.

L262: Is this equation used, and connected to the data?

L266-267: "which can indicate that gravel soil also has the similar principle that the soil with the same grade will shear to reach the same critical void ratio.": But, q and p' differ substantially, explain why.

L269: "The fitting curve": Mainly the fit curves are presented, almost no mechanical and process explanations.

L282-287: Not clear why. Also improve Eng.

L291-292: Fig. 12: What is the difference between filled dots, and open triangles? Also, there is no correlation between them. I don't see the validity of extrapolation. Otherwise, explain these aspects.

L296-298: Does not follow, not clear.

---

## Author Comment (AC1) · 15 Jun 2018

Dear Editors and Reviewers,

We are very grateful to your help and the comments for the manuscript entitled "Mechanical State of Gravel Soil in Mobilization of Rainfall-Induced Landslide in Wenchuan seismic area, Sichuan province, China". Your valuable comments can effectively help our paper improve. We have revised the manuscript in accordance with your detailed comments. Besides, we have carefully proof-read the manuscript to remove mistakes about language and grammar. Please find the following responses to the comments of reviewer. Best wishes. Liping Liao and behalf of all co-authors

[Figure]

Reviewer 1 General comments This paper presented a study about the mechanical state of gravel soil in the landslide initiation using artificial flume model tests and triaxial tests. This topic is very interesting and significant for the landslide early identification and prediction, and it is within the scope of ESURF. The experiment and testing are designed reasonably and its results are reliable. but I think the innovation of this paper is slightly weak. The Introduction and Conclusion did not prepare well. In addition, the language of this paper should be improved. I think this paper needs a round of major revision before publication. Authors' response: Thank you for your kind suggestions. The introduction and conclusion has been rewritten. The language of the manuscript has been improved. The revised details can be found in Line 31∼77, Line 382-397.

Specific comments 1. I think the introduction was not prepared well. too many previous studies were presented, only important studies related to you study should be presented; the purpose and motivation of this paper should be clearer. Authors' response: Thanks a lot for your comment. Your comment provides the valuable guidance for improving the manuscript. According to your suggestion, the introduction has been rewritten and improved. The revised details can be found in Line31-77.

2. The initial dry density is important for the analysis and conclusions, I suggest the authors add some explanation that why or how these four initial dry densities (1.54g/cm3,1.63g/cm3, 1.72g/cm3, 1.81g/cm3) were selected? Authors' response: Thanks you for your comment. The designed initial dry density is 1.50g/cm3,1.60g/cm3, 1.70g/cm3 and 1.80g/cm3. In order to achieve a predetermined density, the soils of the models are divided into four layers, and each layer is compacted respectively. Therefore, some experiment errors exist; the actual density is 1.54g/cm3,1.63g/cm3, 1.72g/cm3 and 1.81g/cm3. The revised details can be found in Line103∼109.

3. In the Section of 3.1, the authors stated that 'throughout the rainfall, the volume moisture content of soil depth of 40cm exhibits a slow-growth trend or remains the

stable'; however, as shown in Fig. 6, the volume moisture content of soil depth of 40cm increased sharply, please provide a brief explanation for this phenomenon. Authors' response: Thanks a lot for your comment. The reason of the phenomenon "the volume moisture content of soil depth of 40cm increased sharply" is its x-label is shorter than x-label of other figures. Therefore, all panels have been plotted for the same x- and y-labels for better comparison. The description about volume moisture content and pore water pressure has been modified accordingly. The revised details can be found in Section 3.2.

4. The authors design the experiment to explore the relationship between the initial dry density and landslide initiation. With the results, it was proved that they have a very close relationship. But, it is still not clear that what the relationship is. For example, why the initiating time of the landslide with the initial dry density of 1.72g/cm3 (18 minutes) is shorter than the landslide with the initial dry density of 1.54-1.63g/cm3 (30 40 minutes). A deep analysis is needed. Authors' response: Thank you for your kind suggestion. A deep analysis on the relationship between initial dry density and landslide initiation has been added. The revised details can be found in Section 3.1. The revised details can be found in Line 165-187.

5. In the Section of Critical state of gravel soil, the gravel soil with an initial dry density of 1.94g/cm3 and 2.00g/cm3 were used, why not the soil sample used before (1.54-1.81g/cm3)? Authors' response: Thank you for your kind suggestion. The one reason is that according to the research (Gabet and Mudd 2006; Iverson et al., 2000), the soil with the same granular composition can obtain the approximate critical void ratio in the uniform stress condition. The other reason is that the authors tried to make the soil sample with 1.54-1.72g/cm3, but the soil sample could not maintain stable when it suffers from the gravity of axial loading system. Based on the above reasons, the density of the soil sample for trixial test is 1.94g/cm3 and 2.00g/cm3.

6. In my opinion, the conclusion section was not written well. the 5th conclusion is not clear; I suggest the conclusions about the Critical state of gravel soil can be synthesized. Technical corrections Line 36-41: please cite only the important references, it is unnecessary to list all the related literature; Line 91: I suggest the authors provide a location map with Niujuan Valley and Duwen highway. Line 93: please check the unit of '32.7Line 116: what does 'CAS' mean, please provide its definition. Line 120: what does 'DL2e' mean; Line 166-167: please correct the sentence; Line 240-241: please check the langue; Tab 2: please check the value of initial dry density, 1.62 or 1.63? it is not clear the meaning of h(cm), soil depth? Tab 3: it is not clear the meaning ofâ ËÏ-UÂ ÿsP0.0075, â ËÏUÂ ÿsP5, â ËÏUÂ ÿsP2 and h. Tab 4: please provide the definition of $\sigma$3; Fig.7-9: please add captions for each subfigure.

(1) In my opinion, the conclusion section was not written well. the 5th conclusion is not clear; I suggest the conclusions about the Critical state of gravel soil can be synthesized. Authors' response: Thanks a lot for your kind suggestion. The conclusion has been rewritten. The revised details can be found in Line 382-397.

(2) Line 36-41: please cite only the important references, it is unnecessary to list all the related literature. Authors' response: Thanks a lot for your kind suggestion. The unimportant references have been removed. The revised details can be found in Line 37.

(3) Line 91: I suggest the authors provide a location map with Niujuan Valley and Duwen highway. Authors' response: Thanks a lot for your kind suggestion. The location map of the study area was provided. The revised details can be found in Line 89.

(4) Line 93: please check the unit of '32.7 Authors' response: Thanks a lot for your kind suggestion. The unit of '32.7 has been checked. 32.7% is the gradient of valley bed, which is equal to the ratio of the height and the length of the valley. So this value is dimensionless.

(5) Line 116: what does 'CAS' mean, please provide its definition. Authors' response: Thank you for your comment. CAS is the abbreviation of Chinese Academy Science. Its definition has been added to Line 110, Line 123.

(6) Line 120: what does 'DL2e' mean; Authors' response: Thank you for your comment. DL2e is the model of the data acquisition system. The revised details can be found in Line 117-119.

(7) Line 166-167: please correct the sentence; Authors' response: Thank you for your comment. The sentence has been corrected. The revised details can be found in Line 252~267.

(8) Line 240-241: please check the langue; Authors' response: Thank you for your comment. The language of Line 240-241 has been modified. The revised details can be found in Line 305~308.

(9) Tab 2: please check the value of initial dry density, 1.62 or 1.63? it is not clear the meaning of h(cm), soil depth? Authors' response: Thank you for your comment. 1.63 is the correct value. The value of initial dry density in Tab 2 has been modified. h is the soil depth and its meaning has been added to Tab.2 and Fig.2(a)

(10) Tab 3: it is not clear the meaning ofâ ËÌUÂÿsP0.0075, â ËÌUÂÿsP5, â ËÌUÂÿsP2 and h. Authors' response: Thanks a lot for your kind suggestion. The cumulative content of coarse (particle diameter > 5mm) is represented by P5, the cumulative content of gravel (particle diameter < 2mm) is represented by P2, and the cumulative content of silt and clay (particle diameter < 0.075mm) is represented by P0.075. The meanings of P5, P2 and P0.075 have been given in section 2.2.3. The revised details can be found in Line 139~141.

(11) Tab 4: please provide the definition of $\sigma3$; Authors' response: Thanks a lot for your kind suggestion. The definition of $\sigma3$ has been added to Tab.4.

(12) Fig.7-9: please add captions for each subfigure; Authors' response: Thanks a lot for your kind suggestion. Due to the adjustment of the structure of Section 3, the figure numbers have been changed. For example, Fig.7-9 is changed to Fig.4~7. The captions of each sub-figures of Fig.4~Fig.7 have been added to the manuscript. The

revised details can be found in Line 207∼223.

References Gabet, E. J.and Mudd, S. M.: The mobilization of debris flows from shallow landslides. Geomorphology, 74, 207-218, doi: 10.1016/j.geomorph.2005.08.013, 2006. Iverson, R. M., Reid, M. E., Iverson, N. R., LaHusen, R. G.and Logan, M.: Acute sensitivity of landslide rates to initial soil porosity. Science, 290, 513-516, doi: 10.1126/science.290.5491.513, 2000.

Please also note the supplement to this comment:
https://www.earth-surf-dynam-discuss.net/esurf-2018-15/esurf-2018-15-AC1-supplement.pdf

---

## Author Comment (AC2) · 15 Jun 2018

Dear Editors and Reviewers,

We are very grateful to your help and the comments for the manuscript entitled "Mechanical State of Gravel Soil in Mobilization of Rainfall-Induced Landslide in Wenchuan seismic area, Sichuan province, China". Your valuable comments can effectively help our paper improve. We have revised the manuscript in accordance with your detailed comments. Besides, we have carefully proof-read the manuscript to remove mistakes about language and grammar. Please find the following responses to the comments of reviewer. Best wishes. Liping Liao and behalf of all co-authors

[Figure]

Reviewer 2 With flume and triaxial tests, this paper investigates the mechanical state of gravel soil in Niujuan valley, Sichuan, China. The authors mentioned that they observed the variation is soil moisture content and pore water pressure, and the macro-micro property. They said to have presented a mathematical expression of critical state of soil. And finally discuss the mechanical state of gravel soil. The topic is very interesting.

However, no new mathematical formulation and model appeared in the text, except for some regression fits. There are several inconsistent statements. Lots of data are presented, great job! But, with much less insights and implications. Both the quality of science and presentation is poor. About 1/2 of the MS is very quantitative and geotechnical, while another 1/2 is very descriptive. How do you relate these data to field events? What are the implications for the surface flow process and run-out modelling? These are not very strongly connected. Large part of the manuscript would perhaps better fit to some geotechnical and civil engineering journals than E-Surf. E.g.L137-153; L191-312. Probably these data would be interesting more to geotechnicians, and perhaps less to the audience of earth surface process. Otherwise, strongly justify how this is not the case. The journal and the Editors can decide on it.

Authors' response: Thank you for your comment. Your comment provides the valuable guidance for improving the manuscript. According to your suggestions, the inconsistent statements have been removed; several sections of the manuscript have been rewritten; the mechanical insights have been added to the manuscript.

The font size is too small. It was very difficult for me to read the print even with the power glasses. Time to time there are > 25 citations at a place! What is the use/purpose of this? This is fully distracting! Why don't you properly utilize the space for useful science/research? I thought the Journal/Editor should also have some initial controls on these and other aspects, at least the basic quality and content of the manuscript, before it is sent for reviews.

[Figure]

Authors' response: Thank you for your comment. The citations have been reduced to 3 citations.

English, in general is good, but time to time difficult to follow, often strange, and needs to be substantially improved.

Authors' response: Thank you for your comment. The languages of the manuscript have been improved.

Detailed and critical comments: L23: "state parameter ...": The audience would not know this here without explaining what they are.

Authors' response: Thank you for your comment. The meaning of state parameter has been added to the introduction. The revised details can be found in Line 25.

L26: "forecast": It is not clear, also in the main text, how you could forcast, what does it mean? Can you predict cracks formation and propagation, time, location and scale for forcasting and warning? No method is presented for this. If possible, please explain clearly how you could do that with the data and the models you are discussing.

Authors' response: Thank you for your comment. The introduction of the manuscript has been revised.

L31,32: Improve English (ENG.). E.g., were locating –> were located, etc.

Authors' response: Thank a lot for your kind suggestion. The language has been improved. The revised details can be found in Line 32-34, Line79.

L36-41: There are > 25 citations here! What is the use/purpose of this? I would suggest to reduce it to about 3.

Authors' response: Thank a lot for your kind suggestion. The citations have been reduced to 3 citations. The revised details can be found in Line 37.

L42: "Fully understanding": Never possible. Improve writing.

Authors' response: Thank a lot for your kind suggestion. This sentence has been rewritten. The revised details can be found in Line 40-41.

L42-46: Looks like introductory undergraduate text.

Authors' response: Thank you for your comment. The introduction of the manuscript has been rewritten. The revised details can be found in Line 38-63.

L50: "Some of the observed phenomena of landslides": Not clear which?

Authors' response: Thank you for your comment. The observed phenomena of landslides included the Salmon Creek landslide in Marin County (Fleming et al., 1989), Slumgullion landslide in Colorado (Schulz et al., 2009), and Guangming New Distinct landslide in Shenzhen (Liang et al., 2017). The revised details can be found in Line 50-52.

L53-55: Again, so may citations. Do you need all these at once? Limit to about 3.

Authors' response: Thank you for your kind suggestion. The citations have been reduced to 3 citations. The revised details can be found in Line 49-50.

L57: Readers would under at this point what F is?

Authors' response: Thank you for your kind suggestion. The F line was drawn by Casagrande (Casagrande A 1936) to distinguish the dilative zone and the contractive zone. This line's horizontal and vertical coordinate is effective normal stress and void ratio. The meaning of F line has been added to Line 47-48.

L59: "the intermittent debris flow": what is it?

Authors' response: Thank you for your comment. The statement of this sentence has been improved. The revised details can be found in Line 55.

L60-69: Strange writing. Unnecessary details, some irrelevant, not connected.

Authors' response: Thank you for your comment. Unnecessary details have been removed. In addition, the introduction of the manuscript has been rewritten. The revised details can be found in Line 38-64.

L74: "landslide velocity": Which velocity? Initiation, or dynamical until runout? You did not present data and analysis for velocity. Also, the dynamic velocity would, at most, negligibly depend on the initial state you are referring to. Otherwise, present data and analysis to support your arguments.

Authors' response: Thank you for your comment. The statement was provided by William (Schulz et al., 2009). He pointed out the dilative strengthening might control the velocity of a moving landslide though the hourly continuous measurement of displacement of landslide. Therefore, "landslide velocity" is the velocity of the dynamic movement of landslide. The revised details can be found in Line 59-60.

L75-80: Again, > 25 citations at one place. This is fully distracting! Why don't you properly utilize the space for useful science/research?

Authors' response: Thank you for your kind suggestion. The citations have been reduced. The revised details can be found in Line 63-67.

L81-80: "the critical state of gravel soil in a seismic area is not exactly identified in the field research": Why does it matter if it is sesimic or not?

Authors' response: Thank a lot for your comment. Gravel soils are generated by seismic shaking in Wenchuan earthquake area (Tang and Liang 2008; Xie et al., 2009). The feature of this soil is wide grading, under-consolidation and low density. In addition, according to the existing literatures, the research on the critical state of gravel soil is lacking at present. Therefore, this study is necessary and has the local characteristic.

L96, 102: "large scale", "most of rainfall induced landslides is the shallow landslides": inconsistent presentations. What is large scale?

Authors' response: Thank you for your kind suggestion. According to the field investigations, debris flow is large scale. So the statement has been improved. The revised

details can be found in Line 86.

L103-104: "silt and clay (particle diameter < 0.075mm) is about 2\%, which plays the important role in the mobilization of landslide and debris flow": How? Without proof and discussion, statements are useless.

Authors' response: Thank you for your kind suggestion. Chen (Chen et al., 2010) provided the valuable evidence for quantifying clay content impact on gravel soil failure and the initiation of debris flow. He concluded that silt and clay content played the important role in the mobilization of landslide and debris flow. Therefore, authors only cited his conclusion in the manuscript. The revised details can be found in Line 100-102.

L119: "produced in England": Do you need to say this? Why not to use reference properly?

Authors' response: Thank you for your kind suggestion. The unnecessary information "produced in England" has been removed. The revised details can be found in Line 116-118.

L127-129: Fig. 1a: Initial shape and wedge angle needs to be discussed, also why chosen this way?

Authors' response: Thank you for your kind suggestion. The reasons for choosing initial conditions of test have been added to Section 2.2.1. The revised details can be found in Line 95, Line 102-109, Line 113-115.

L143: "The mean effective stress p' is equal to one third of the sum of $\sigma x$, $\sigma y$ and $\sigma z$":Do you really need to say this? There are lots of unnecessary things, making the MS much less professional.

Authors' response: Thank you for your kind suggestion. The statement of these problems has been revised in this manuscript. In addition, although this sentence represents the traditional theory of soil mechanics, it is also useful for the manuscript

because p' is an important parameter of the soil state, which represents the stress condition of a certain point in the artificial flume model. If the formula of p' is not stated in the manuscript, the reader cannot understand Table 2. The revised details can be found in Line 141.

L145: "is the soil bulk density": No!

Authors' response: Thank you for your comment.$\gamma$ is the unit weight of soil. The definition of $\gamma$ has been modified. The revised details can be found in Line 143.

L156-160: Eng.

Authors' response: Thank you for your kind suggestion. Section 3.2 has been rewritten. The revised details can be found in Line 226-268.

L168-175: The yellow lines in Fig. panels cannot be seen. Better, plot in different line styles. Explain why the yellow lines are mostly in between the other lines on the right panels? All panels must be plot for the same x- and y-labels for better comparison. The mechanical and geotechnical reasons for the spacial behaviors seen in these panels are not well explained. Furthermore, how these behaviors influence dilation, landslide initiation, velocity and run-out?

Authors' response: Thank you for your kind suggestions. (1) The line styles have been modified and all panels have been plotted for the same x- and y-labels. The revised details can be found in Line 268-275. (2) The reason for the yellow line's location had been added to section 3.2. The mechanical and geotechnical reasons for the spacial behaviors seen in these figures were explained. The revised details can be found in Line 226-267. (3) The influence of volume moisture content and pore water pressure on dilation, landslide initiation has been added to section 4. The revised details can be found in Line 364-380.

L178: "the landslide can be triggered by rainfall": Show the hydro-mechanical relationship with the above figure. Otherwise, what is the use of the above data?

Authors' response: Thank you for your kind suggestion. A camera was used to record the macroscopic process of the entire experiment. Thus landslide triggered by rainfall was the phenomenon of the model tests. In addition, the hydro-mechanical relationship with the above figures had been added to Section 3.2.

L184-185: Eng.

Authors' response: Thank you for your kind suggestion. The language has been improved. The revised details can be found in Line 159-164.

L185-186: "For example, when the initial dry density is 1.54âĹij1.63g/cm3, the initiating time of landslide is 30âĹij40 minutes": You must relate this with Fig. 6, right panels. No insight about the mechanics and process are mentioned, linked, and discussed. Otherwise, what is the use of Fig. 6?

Authors' response: Thank you for your kind suggestion. The differences between Fig.8∼Fig.11 has been added to Section 3.2 (Line 226-267). The mechanics and process linking with these figures have been added to Section 3.1 and 3.2.

L191: "expansion of cracks": Show it and the dynamics.

Authors' response: Thank you for your kind suggestion. Fig.6 (c) has been added to show the propagation of cracks. The revised details can be found in Line 218.

L192: "and rotation": how, where do you see it?

Authors' response: Thank you for your kind suggestion. This phenomenon is not my observation, but is observed by other researchers (Gao et al., 2011; Igwe 2014). The relative references have been citied. The revised details can be found in Line 160.

L193-194: "All the above process can lead to the decrease of the void ratio and the increase of the pore water pressure": Not clear how?

Authors' response: Thank you for your kind suggestion. This statement has been improved. The revised details can be found in Line 158-164.

L195-196: "When the initial dry density is 1.81g/cm 3, the slope keeps stable and landslide cannot be triggered by the rainfall even though the fine particles disappear, and the coarse particles are exposed at the slope surface.": This is important. Explain with strength relation.

Authors' response: Thank you for your kind suggestion. The reasons for this phenomenon have been added. The revised details can be found in Line 191-197.

L196-205: The figure captions don't explain the process in panels, difficult to follow.

Authors' response: Thank you for your kind suggestion. The captions for each sub-figure of Fig.4-Fig.7 have been added. The revised details can be found in Line 201-223.

L252-254: Difficult to follow.

Authors' response: Thank you for your kind suggestion. The definition of critical state has been improved. The revised details can be found in Line 319-323.

L262: Is this equation used, and connected to the data?

Authors' response: Thank you for your kind suggestion. The formula (2) was used to calculate the critical void ratio. The revised details can be found in Line 323-326.

L266-267: "which can indicate that gravel soil also has the similar principle that the soil with the same grade will shear to reach the same critical void ratio.": But, q and p' differ substantially, explain why.

Authors' response: Thank you for your comment. This principle is from the "critical state soil mechanics" (Casagrande A 1936; Roscoe et al., 1963; Schofield and Wroth 1968), which has been validated by many researchers (Fleming et al., 1989; Gabet and Mudd 2006; Iverson et al., 2000)). The revised details can be found in Line 331-332.

L269: "The fitting curve": Mainly the fit curves are presented, almost no mechanical and process explanations.

Authors' response: Thank you for your kind suggestion. The mechanical meaning of the fitting curve has been added. The revised details can be found in Line 341-344.

L282-287: Not clear why. Also improve Eng.

Authors' response: Thank you for your kind suggestion. Section 4 has been rewritten. The revised details can be found in Line 354-361.

L291-292: Fig. 12: What is the difference between filled dots, and open triangles? Also, there is no correlation between them. I don't see the validity of extrapolation. Otherwise, explain these aspects.

Authors' response: Thank you for your kind suggestion. Six filled dots represent the critical state of soil; their values, including ec and lnp', can be derived from triaxial tests (Tab.4). In addition, the critical state line is obtained by fitting these values (Line 335-339). The hollow dots represent the current states of the soils; the state parameters (e, p') can be derived from the artificial flume model tests (Tab.2). These dots have a close correlation. The critical state line can divide the graphical space into two states. The space above this curve is the contractive zone, and the space below this curve is the dilative zone. If the state parameter (e, p') is determined, the soil state can be judged by this line (Gabet and Mudd 2006; Iverson et al., 2000). Therefore, the mechanical state of soil in the artificial flume model can be determined according to Fig.14. Although there are three confining pressures in triaxial tests, the fitting curve of ec and lnp' still has a significant statistical meaning due to its high correlation coefficient. In future, multiple confining pressures will be considered in tests to validate the extrapolation of this curve.

L296-298: Does not follow, not clear.

Authors' response: Thank you for your kind suggestion. Section 4 has been rewritten. The revised details can be found in Line 364-380.

References

Casagrande A: Characteristics of cohesionless soils affecting the stability of slopes and earth fills. Journal of the Boston Society of Civil Engineers, 23, 13-32, 1936.

Chen, N. S., Zhou, W., Yang, C. L., Hu, G. S., Gao, Y. C.and Han, D.: The processes and mechanism of failure and debris flow initiation for gravel soil with different clay content. Geomorphology, 121, 222-230, doi: 10.1016/j.geomorph.2010.04.017, 2010.

Fleming, R. W., Ellen, S. D.and Algus, M. A.: Transformation of dilative and contractive landslide debris into debris flows-An example from marin County, California. Engineering Geology, 27, 201-223, 1989.

Gabet, E. J.and Mudd, S. M.: The mobilization of debris flows from shallow landslides. Geomorphology, 74, 207-218, doi: 10.1016/j.geomorph.2005.08.013, 2006.

Gao, B., Zhou, J.and Zhang, J.: Macro-meso analysis of water-soil interaction mechanism of debris flow starting process. Chinese Journal of Rock Mechanics and Engineering, 30, 2567-2573, 2011 (in Chinese).

Igwe, O.: The compressibility and shear characteristics of soils associated with landslides in geologically different localities—case examples from Nigeria. Arabian Journal of Geosciences, 8, 6075-6084, doi: 10.1007/s12517-014-1616-3, 2014.

Iverson, R. M., Reid, M. E., Iverson, N. R., LaHusen, R. G.and Logan, M.: Acute sensitivity of landslide rates to initial soil porosity. Science, 290, 513-516, doi: 10.1126/science.290.5491.513, 2000.

Liang, H., He, S. m., Lei, X. q., Bi, Y. z., Liu, W.and Ouyang, C. j.: Dynamic process simulation of construction solid waste (CSW) landfill landslide based on SPH considering dilatancy effects. Bulletin of Engineering Geology and the Environment, 2, 1-15, doi: 10.1007/s10064-017-1129-x, 2017.

Roscoe, K. H., Schofield, A. N.and Thuraijajah, A.: Yielding of clays in states wetter

than critical. Geotechnique, 13, 211-240, 1963.

Schofield, A. N.and Wroth, C. P. Critical state soil mechanics. University of Cambridge, 1968.

Schulz, W. H., McKenna, J. P., Kibler, J. D.and Biavati, G.: Relations between hydrology and velocity of a continuously moving landslide - evidence of pore-pressure feedback regulating landslide motion? Landslides, 6, 181-190, doi: 10.1007/s10346-009-0157-4, 2009.

Tang, C.and Liang, J. T.: Characteristics of debris flows in Beichuan epicenter of the Wenchuan earthquake triggered by rainstorm on september 24, 2008. Journal of Engineering Geology, 16, 751-758 (in Chinese), doi: 10.1016/j.geomorph.2005.08.013, 2008.

Xie, H., Zhong, D. L., Jiao, Z.and Zhang, J. S.: Debris flow in Wenchuan quake-hit area in 2008. Mountain Research, 27, 501-509, 2009 (in Chinese).

Please also note the supplement to this comment:
https://www.earth-surf-dynam-discuss.net/esurf-2018-15/esurf-2018-15-AC2-supplement.pdf